# PIXEL MOTION DIFFUSION IS WHAT WE NEED FOR ROBOT CONTROL

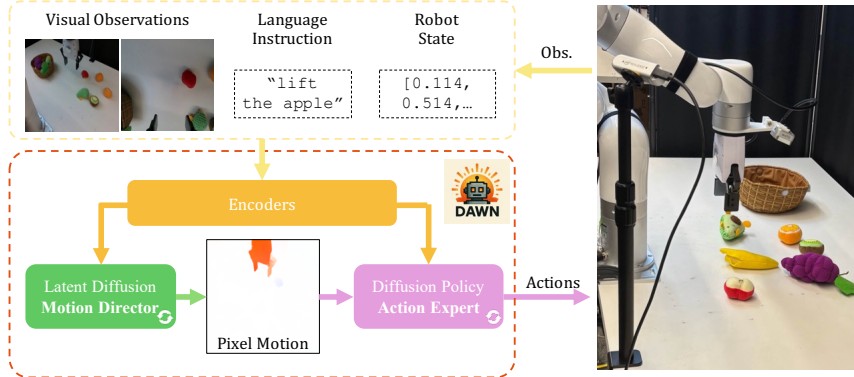

Figure 1: Overview of DAWN with two major diffusion modules. First, observations are encoded into conditional embeddings; Based on that, a latent diffusion Motion Director generates a pixel motion representation, which the diffusion policy Action Expert uses to create robot actions.

## ABSTRACT

We present DAWN (Diffusion is All We Need for robot control), a unified diffusion-based framework for language-conditioned robotic manipulation that bridges high-level motion intent and low-level robot action via structured pixel motion representation. In DAWN, both the high-level and low-level controllers are modeled as diffusion processes, yielding a fully trainable, end-to-end system with interpretable intermediate motion abstractions. DAWN achieves state-of-the-art results on the challenging CALVIN benchmark, demonstrating strong multi-task performance, and further validates its effectiveness on MetaWorld. Despite the substantial domain gap between simulation and reality and limited real-world data, we demonstrate reliable real-world transfer with only minimal finetuning, illustrating the practical viability of diffusion-based motion abstractions for robotic control. Our results show the effectiveness of combining diffusion modeling with motion-centric representations as a strong baseline for scalable and robust robot learning. Visualizations at `anonymous.4open.science/w/DAWN`.

## 1 INTRODUCTION

Multi-stage pixel or point tracking based methods have recently emerged as a promising direction for robot manipulation, offering interpretable intermediate pixel motion and modular control (Yuan et al., 2024a; Gao et al., 2024; Xu et al., 2024; Bharadhwaj et al., 2024b;a; Ranasinghe et al., 2025). However, despite their promise, approaches such as Im2Flow2Act (Xu et al., 2024), ATM (Wen et al., 2023), and LangToMo (Ranasinghe et al., 2025) still fall short of state-of-the-art vision-language action (VLA) models (Black et al., 2024a; Intelligence et al., 2025) and latent feature–based hierarchical methods (Hu et al., 2024; Nvidia et al., 2025) on established benchmarks. We argue that this performance gap does not arise from limitations in the two-stage intermediate pixel-motion based framework itself. The high-level motion generator in these frameworks does not fully reflect recent advances in visual generative modeling (Ge et al., 2022; Kumari et al., 2023; Zhang et al., 2022; Ren et al., 2022; Chen et al., 2023), while the low-level controllers have not

leveraged recent progress in diffusion-based action policies (Janner et al., 2022; Du et al., 2023a; Chi et al., 2023; Shridhar et al., 2024; Li et al., 2024a) in an optimal way.

To address these limitations, we introduce a two-stage diffusion-based visuomotor framework in which both the high-level and low-level controllers are instantiated as diffusion models and glued by explicit pixel motions as illustrated in Figure 1. The high-level motion director, which is a latent diffusion module, takes current (multiview) visual observations and language instruction, and predicts desired dense pixel motion from a third-person view. This pixel motion could be regarded as a structured intermediate representation of desired scene dynamics to accomplish the language instruction. These pixel motion are then translated into executable actions through a diffusion-based policy head. We highlight how intermediate pixel motion is grounded on visual inputs, endowing the intermediate representations with interpretability. Therein, we introduce **D**iffusion is **A**ll **W**e **N**eed for robot control (DAWN), which bridges the strengths of hierarchical motion decomposition and end-to-end visuomotor agents, while maintaining interpretability and modularity.

Our framework illustrated in Figure 1 builds upon insights from prior hierarchical visuomotor approaches. VPP (Hu et al., 2024) employs a video diffusion model to extract predictive feature embeddings, which subsequently condition a downstream action policy. However, it operates in RGB space (with no motion specific representation) and uses the video diffusion model as a feature extractor as opposed to iterative denoising of motion features. LangToMo (Ranasinghe et al., 2025) predicts pixel-space motion trajectories from language instructions, but its high-level motion director uses pixel-level diffusion, limiting the resolution of the generated motion representation and training scalability. Its low-level controller is based on weaker ViT architectures or hand-crafted heuristics. In contrast, DAWN utilizes an efficient pretrained latent diffusion model for motion generation with iterative denoising during inference, and a strong diffusion-based action expert, thus benefiting from powerful vision and language models.

We evaluate our method on two challenging simulation benchmarks—CALVIN (Mees et al., 2022) and MetaWorld (Yu et al., 2019), as well as across real-world environments with only very limited in-domain training data.

Our results demonstrate that, despite using limited data and substantially smaller model capacity, our method can match or even surpass state-of-the-art VLA models by leveraging explicit structured pixel motion and the strengths of diverse pretrained models, highlighting its high data efficiency.

Our key contributions are as follows:

1. We propose DAWN, a two-stage diffusion-based framework that generates structured intermediate pixel motion as an efficient language-conditioned visuomotor policy.

2. Despite relying on limited data and a substantially smaller model capacity, we achieve competitive or even state-of-the-art performance on CALVIN, MetaWorld, and real-world benchmarks.

3. Our approach is explicitly designed to leverage pretrained vision and language models, enabling highly data-efficient transfer across domains, while providing interpretability and modularity.

## 2 RELATED WORK

**Pixel Motion for Robot Control:** Several prior works explore pixel trajectories or optical flow as motion representations (Bharadhwaj et al., 2024b;a; Hu et al., 2024; Ranasinghe et al., 2025). These methods capture the displacement of pixels between consecutive frames, providing a dense and local description of motion that is universal and often embodiment-agnostic. Recent advances have leveraged these representations to enable scalable robot learning and zero-shot skill transfer. For instance, LangToMo (Ranasinghe et al., 2025) introduces a dual-system framework that uses language-conditioned pixel motion forecasts as an intermediate representation, allowing robot control to be learned from web-scale video-caption data without requiring specific robot action annotations. Similarly, General Flow (Yuan et al., 2024a) proposes a language-conditioned 3D flow prediction model trained on large-scale human videos and treats 3D flow as a foundational affordance, providing a scalable, universal language for describing manipulation.

Other works focus on using pixel motion for planning and policy learning. FLIP (Gao et al., 2024) utilizes a flow-centric generative planning model to synthesize long-horizon plans from

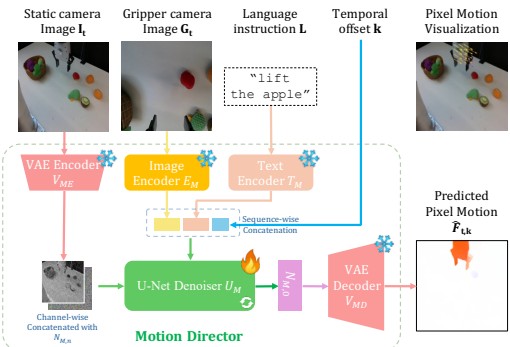 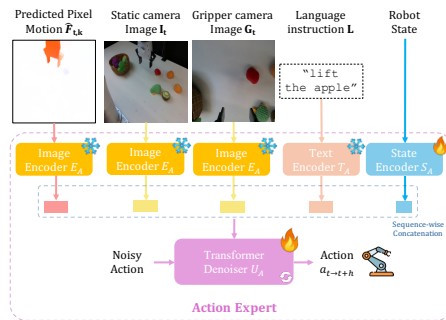

Figure 2: Architecture of Motion Director. The model encodes the static camera view and denoises it with a U-Net, conditioned on the gripper view, language instruction with a temporal offset. The output is decoded into predicted pixel motions, providing interpretable motion representations.

Figure 3: Architecture of Action Expert. The model encodes predicted pixel motion, visual observations, language instruction, and robot state into multimodal features. These inputs condition the denoising process, which iteratively refines noisy actions into executable robot trajectories.

language-annotated videos, guiding low-level policies. Im2Flow2Act (Xu et al., 2024) and Track2Act (Bharadhwaj et al., 2024b) both use point or object flow as a cross-domain interface, bridging the gap between human videos, simulated data, and real-world robot execution to achieve zero-shot manipulation. Finally, Gen2Act (Bharadhwaj et al., 2024a) takes a generative approach, first imagining a video of future motion in image pixel space and then conditioning a robot policy on the generated video to enable generalizable manipulation.

**Vision-Language-Action Models with Pixel-related Representations:** Vision-language-action models have emerged as a powerful paradigm for language-conditioned robot control (Brohan et al., 2023; Brohan & et al., 2023; Bahl et al., 2022; Padalkar & et al., 2023; Reed et al., 2022; Wu et al., 2023; Driess et al., 2023; Kim et al., 2024; Zheng et al., 2024; Zawalski et al., 2024; Sudhakar et al., 2024; Jeong et al., 2025; Yang et al., 2025). Leveraging large-scale training with web-scale vision-language data, these models increasingly focus on improving generalization and data efficiency. DVD (Chen et al., 2021) uses diverse "in-the-wild" human videos to teach reward functions and enables zero-shot transfer to new environments. Other approaches focus on learning from passive observation, as seen in (Ko et al., 2023), who developed a policy that learns from "actionless" videos by inferring actions from dense correspondences between generated future frames. GR-1 (Wu et al., 2023) is a GPT-style transformer policy that benefits from large-scale video pre-training. 3D-VLA (Zhen et al., 2024) proposes a world model that integrates 3D perception and reasoning to enhance planning capabilities. These advances have been supported by initiatives like Octo (Octo Model Team et al., 2024), an open-source generalist policy trained on the vast Open X-Embodiment dataset (O'Neill et al., 2024), paving the way for more reproducible and widely usable models.

Pixel-related representations are also found useful in robot manipulation. GENIMA (Shridhar et al., 2024) fine-tunes a diffusion model to inpaint markers on visual observations, which could be decoded into robot actions. LLaRA (Li et al., 2024b) presents the robot action in text-based image pixel coordinates and formats the robot policy into a conversation style to benefit from a pretrained large VLM. This enables an efficient transfer from a general VLM into VLA. Similarly, RoboPoint (Yuan et al., 2024b), LLARVA (Niu et al., 2024), TraceVLA (Zheng et al., 2024) and Magma (Yang et al., 2025) all take advantage of different kinds of image coordinate-based representations.

## 3 METHOD

### 3.1 PRELIMINARIES: DIFFUSION MODELS

Diffusion models (Sohl-Dickstein et al., 2015; Ho et al., 2020; Song et al., 2021) are powerful generative models that synthesize data by iteratively denoising noise-corrupted inputs to approximate the target data distribution. The process involves a forward step that gradually perturbs real data with noise through a Markov chain, and a reverse step in which a neural network parameterizes Gaussian transitions to progressively remove noise, eventually generating realistic samples from a

simple distribution such as a standard Gaussian. These models have shown remarkable success in image generation and broader data generation tasks. To improve scalability and efficiency for image generation, latent diffusion models (Rombach et al., 2022) operate in a compressed latent space rather than raw pixel space, significantly reducing computational demands while preserving fidelity.

Diffusion-based approaches have also been adapted for robot learning, where policy learning can be framed as a sequence generation problem. Diffusion Policy (Chi et al., 2023), in particular, addresses the challenge of visuomotor control by generating action sequences conditioned on both visual and low-dimensional states.

### 3.2 Problem Formulation and DAWN Overview

We study the topic of language-instructed visuomotor control, where the goal is to build a policy that takes both visual observations and natural language instructions from the environment to generate robot actions for control, using behavior cloning (i.e., supervised learning).

Our approach, DAWN, combines the strengths of two complementary diffusion models: a latent image diffusion model for pixel-level motion generation, referred to as *Motion Director*, and a diffusion transformer for fine-grained action sequence generation, referred to as *Action Expert*. These two models interact through explicit pixel-motion representations. At a higher level, Motion Director conditions on multi-view images and the language instruction to iteratively generate task-aligned pixel motions, grounded to one of the input views as illustrated at Figure 2. At the lower level, Action Expert takes the generated pixel motion along with additional inputs to produce the final robot action sequence as illustrated at Figure 3. We highlight how our pixel-motions are grounded to an input view, endowing our intermediate representations with interpretability.

### 3.3 Motion Director: Language-to-Motion Generation

Consider two videos $\mathbf{I}, \mathbf{G} \in \mathbb{R}^{T \times H \times W \times C}$ capturing the same robot demonstration from different camera views, each consisting of $T$ frames of height $H$, width $W$, and $C$ channels, along with the corresponding language instruction $\mathbf{L}$. For example, $\mathbf{I}$ could be the video from a static third-person view, and $\mathbf{G}$ could be captured from the camera above the gripper. Let $\mathbf{I}_t, \mathbf{G}_t$ denote the $t$-th frame from the corresponding views. We define the pixel motion from $\mathbf{I}_t$ to $\mathbf{I}_{t+k}$ as $\mathbf{F}_{t,k} = [u, v]$, where $u, v \in \mathbb{R}^{H \times W}$ represent amount of movement of each pixel between $\mathbf{I}_t$ and $\mathbf{I}_{t+k}$ in the horizontal and vertical directions, respectively. To take advantage of pretrained models, we further encode this motion into a three-channel image $\mathbf{F}'_{t,k} = [u, v, (u + v)/2]$.

The goal of Motion Director is to estimate $\mathbf{F}'_{t,k}$ using only current visual input $\mathbf{I}_t, \mathbf{G}_t$ and instruction $\mathbf{L}$. Our Motion Director builds on a pretrained latent diffusion model for RGB image generation, comprising a U-Net denoiser $U_M$, a text encoder $T_M$, and pretrained VAE encoder–decoder pair $(V_{ME}, V_{MD})$. We also incorporate a vision encoder $E_M$ to extract embeddings from alternative camera views.

At the inference time, we first draw a Gaussian noise tensor $\mathbf{N}_{M,n}$ and concatenate it with the latent encoding of the current frame $V_{ME}(\mathbf{I}_t)$, forming a noisy latent representation $\mathbf{O}_{M,n} = [N_{M,n}, V_{ME}(\mathbf{I}_t)]$, where $n$ is the total number of denoising steps we plan to execute. Note that the current frame latent encoding $V_{ME}(\mathbf{I}_t)$ does not undergo any form of corruption, as this is a conditioning signal. The U-Net $U_M$ then denoises $\mathbf{O}_{M,n}$ and outputs a less noisy latent tensor $N_{M,n-1}$ under the conditioning of the language embedding $T_M(\mathbf{L})$, visual embedding of the alternative view $E_M(\mathbf{G}_t)$, and the temporal offset $k$. All conditioning tokens are concatenated and injected into the U-Net's cross-attention layers at each denoising step. The denoised latent tensor will be concatenated again with the VAE encoded visual inputs to form the input for the next denoising step $\mathbf{O}_{M,n-1} = [N_{M,n-1}, V_{ME}(\mathbf{I}_t)]$. For an arbitrary denoising step $i$, the process can be presented as Equation (3) where $t_i$ is the denoising timestamp and [...] stands for concatenation.

$$\mathbf{O}_{M,i} = [N_{M,n}, V_{ME}(\mathbf{I}_t)] \tag{1}$$

$$\mathbf{C}_M = [E_M(\mathbf{G}_t), T_M(\mathbf{L}), k] \tag{2}$$

$$\mathbf{N}_{M,i-1} = U_M(\mathbf{O}_{M,i}, \mathbf{C}_M, t_i) \tag{3}$$

After $n$ iterations, the denoised latent tensor $\mathbf{N}_{M,0}$ is decoded by $V_{MD}$ into a three-channel image, which ideally matches the ground-truth motion $\mathbf{F}'_{t,k}$.

During training, we update only the U-Net denoiser $U_M$, while keeping all other modules frozen. The ground-truth pixel motion corresponding to frame $\mathbf{I}_t$ is obtained using the optical flow model RAFT (Teed & Deng, 2020) since we have access to future frames during training (i.e., using frames $\mathbf{I}_t$ and $\mathbf{I}_{t+k}$ as input to RAFT), and subsequently projected into latent space through the VAE encoder $V_{ME}$.

### 3.4 ACTION EXPERT: DIFFUSION-BASED POLICY

Our Action Expert is responsible for translating pixel motions into low-level robot actions, conditioned on visual observations, robot states, and language instructions. To achieve this, motivated by prior diffusion based policies (Chi et al., 2023), we construct a transformer based Enhanced Diffusion Policy, which generates action sequences by progressively denoising noisy action representations under multimodal conditions. This design enables the policy to capture complex dependencies across modalities while producing coherent actions temporally.

The architecture consists of four key components: (1) a shared visual encoder $V_A$ that encodes both the pixel motion output from Motion Director and the current visual observations, (2) a text encoder $T_A$ that embeds the language instruction, (3) a state encoder $S_A$ that processes low-dimensional robot states through a two-layer MLP, and (4) a denoising transformer $U_A$ that generates action sequences. We initialize $U_A$ and $S_A$ from scratch to allow adaptation to the target task, while keeping the pretrained $V_A$ and $T_A$ frozen to benefit from strong pretrained visual and language representations.

During inference, the pixel motion predicted by Motion Director, together with the visual inputs, language instruction, and robot states, are each processed by their corresponding encoder and projected into token embeddings. These context tokens are concatenated to form the conditioning sequence, which is injected into all transformer blocks of the denoising transformer $U_A$ via cross-attention, following the same mechanism as in Motion Director. Action generation begins from a noisy action chunk with length $h$ sampled from a Gaussian prior, which is iteratively denoised by $U_A$ into a coherent sequence of executable robot actions.

### 3.5 DAWN TRAINING AND INFERENCE

In summary, both Motion Director and Action Expert are trained with a mean squared error noise estimation loss. While Motion Director operates in the latent image space to predict pixel motions, Action Expert focuses on predicting action chunks in the robot's action space.

At inference time, all the observations are first encoded into condition representations. Conditioned on that, Motion Director then iteratively generates a single pixel motion image , which serves as input to Action Expert. Considering this pixel motion and the other representations, Action Expert finally produces a sequence of executable robot actions through a similar recurrent denoising process. Once these actions are executed, the system repeats the process with the updated observations, thereby forming a closed-loop control pipeline.

This hierarchical design leverages the strengths of large pretrained models in both computer vision and robotics, while maintaining modularity and interpretability through the explicit use of pixel motion as an intermediate representation. One advantage of this modularity is that the two diffusion models can be trained in parallel using the optical flow between two images as the groundtruth pixel motion. Two modules could be upgraded independently, enabling flexible integration of future advances in vision or control. After that, Action Expert could optionally be further fine-tuned on the actual pixel motions generated by Motion Director for a better performance.

To the best of our knowledge, this is the first work to adapt a pretrained *latent* diffusion model for dense pixel *motion* generation and use the pixel motion to guide a diffusion policy for visuomotor control under fully learnable settings.

## 4 EXPERIMENTS

We evaluate our framework on two challenging simulation benchmarks—CALVIN (Mees et al., 2022) and MetaWorld (Yu et al., 2019), as well as across real-world environments involving diverse

Table 1: **CALVIN Evaluation (no external robotic data):** Results reported for zero-shot long-horizon evaluation on the Calvin ABC→D benchmark, where the agent is asked to complete five chained tasks sequentially based on instructions. All methods are trained only on the CALVIN dataset without any external data.

| Method | $i^{th}$ Task Success Rate | | | | | Avg. Len ↑ |
|---|---|---|---|---|---|---|
| | 1 | 2 | 3 | 4 | 5 | |
| Diffusion Policy (Chi et al., 2023) | 0.402 | 0.123 | 0.026 | 0.008 | 0.00 | 0.56 |
| Robo-Flamingo (Li et al., 2023) | 0.824 | 0.619 | 0.466 | 0.331 | 0.235 | 2.47 |
| RoboUniview (Yang et al., 2025) | 0.942 | 0.842 | 0.734 | 0.622 | 0.507 | 3.65 |
| Seer (Tian et al., 2024) | 0.930 | 0.824 | 0.723 | 0.626 | 0.533 | 3.64 |
| Seer-Large (Tian et al., 2024) | 0.927 | 0.846 | 0.761 | 0.689 | 0.603 | 3.83 |
| VPP (Hu et al., 2024) | 0.955 | 0.879 | 0.784 | 0.714 | 0.604 | 3.93 |
| Enhanced Diffusion Policy (ours) | 0.824 | 0.672 | 0.528 | 0.408 | 0.352 | 2.78 |
| DAWN (ours) | 0.981 | 0.913 | 0.788 | 0.712 | 0.606 | 4.00 |

Table 2: **CALVIN Evaluation with external robotic data:** Zero-shot long-horizon evaluation on the Calvin ABC→D benchmark where agent is asked to complete five chained tasks sequentially based on instructions.

| Method | Additional Data | $i^{th}$ Task Success Rate | | | | | Avg. Len ↑ |
|---|---|---|---|---|---|---|---|
| | | 1 | 2 | 3 | 4 | 5 | |
| GR-1 (Wu et al., 2023) | Ego4D | 0.854 | 0.712 | 0.596 | 0.497 | 0.401 | 3.06 |
| Vidman (Wen et al., 2024) | OpenX subsets | 0.915 | 0.764 | 0.682 | 0.592 | 0.467 | 3.42 |
| LTM (Ranasinghe et al., 2025) | OpenX subsets | 0.971 | 0.824 | 0.728 | 0.672 | 0.606 | 3.81 |
| Seer (Tian et al., 2024) | DROID | 0.944 | 0.872 | 0.799 | 0.722 | 0.643 | 3.98 |
| Seer-Large (Tian et al., 2024) | DROID | 0.963 | 0.916 | 0.861 | 0.803 | 0.740 | 4.28 |
| VPP (Hu et al., 2024) | Multiple sources | 0.965 | 0.909 | 0.866 | 0.820 | 0.769 | 4.33 |
| DreamVLA (Zhang et al., 2025) | DROID | 0.982 | 0.946 | 0.895 | 0.834 | 0.781 | 4.44 |
| DAWN (ours) | DROID | 0.978 | 0.916 | 0.813 | 0.752 | 0.641 | 4.10 |

robotic manipulation tasks. In this section, we first introduce our experimental setup, followed by evaluations on the three selected robotics environments, and finally ablation studies.

## 4.1 IMPLEMENTATION DETAILS

Our DAWN comprises two components, Motion Director and Action Expert. We initialize our Motion Director from a pretrained latent diffusion model from (Rombach et al., 2022; 2025) that has been trained on large-scale image-text datasets. The additional U-Net weights we use for our additional visual conditioning are zero-initialized to ensure that the pretrained network behavior is preserved at the beginning of training, and the model can gradually adapt to the additional input modality. We encode the language instruction using a pretrained CLIP text encoder, and extract gripper view visual tokens using a CLIP ViT encoder. During the inference, we use 25 diffusion steps to generate the final pixel motion prediction.

Our Action Expert which contains a diffusion policy conditioned on visual, textual, and robotic state modalities uses different encoders for each input. The visual encoder is a pretrained ConvNeXt-S variant of DINOv3 (Siméoni et al., 2025), and the text encoder is a T5-small pretrained model. The state encoder and the diffusion policy head are randomly initialized.

## 4.2 CALVIN EXPERIMENTS

We first evaluate our DAWN on the CALVIN benchmark (Mees et al., 2022). This simulated benchmark measures the long-horizon capability of robotic manipulation tasks. We select this environment for the challenging nature of its tasks, requiring semantic understanding and 3D awareness.

**Dataset:** This benchmark provides a dataset containing 4 different splits, A, B, C, and D, each containing demonstrations from distinct environments. Across scenes, the dataset contains 34 tasks, with a total of 24k demonstrations. We focus on the most challenging ABC→D task setting, where the model is trained on the A, B, and C environments and then evaluated in the unseen D environment. Several prior works also report results using pretraining on external data, including (Hu et al., 2024; Ranasinghe et al., 2025; Zhang et al., 2025; Gu et al., 2023). We train our model under this setting as well, where we use the DROID dataset (Khazatsky et al., 2024) for our pretraining.

**Evaluation:** We follow standard evaluation protocol from (Hu et al., 2024), which evaluates a given policy on 1000 episodes each containing 5 continuous tasks (i.e. task $i$ starts from the end state of task $i-1$, which is often different to what is encountered in demonstrations within the training data).

Table 3: **MetaWorld task success rate:** Our DAWN achieves state-of-the-art performance on MetaWorld.

| Method | door-open | door-close | basketball | shelf-place | btn-press | btn-top | faucet-close | faucet-open | handle-press | hammer | assembly | Overall |
|---|---|---|---|---|---|---|---|---|---|---|---|---|
| BC-Scratch (Nair et al., 2022) | 21.3 | 36.0 | 0.0 | 0.0 | 34.7 | 12.0 | 18.7 | 17.3 | 37.3 | 0.0 | 1.3 | 16.2 |
| BC-R3M (Nair et al., 2022) | 1.3 | 58.7 | 0.0 | 0.0 | 36.0 | 4.0 | 18.7 | 22.7 | 28.0 | 0.0 | 0.0 | 15.4 |
| Diffusion Policy | 45.3 | 45.3 | 8.0 | 0.0 | 40.0 | 18.7 | 22.7 | 58.7 | 21.3 | 4.0 | 1.3 | 24.1 |
| UniPi (Du et al., 2023b) (With Replan) | 0.0 | 36.0 | 0.0 | 0.0 | 6.7 | 0.0 | 4.0 | 9.3 | 13.3 | 4.0 | 0.0 | 6.1 |
| Im2Flow2Act (Xu et al., 2024) | 0.0 | 0.0 | 0.0 | 4.0 | 0.0 | 0.0 | 7.3 | 4.7 | 0.0 | 0.0 | 0.0 | 2.0 |
| ATM (Wen et al., 2023) | 75.3 | 90.7 | 24.0 | 16.3 | 77.3 | 76.7 | 50.0 | 62.7 | 92.3 | 4.3 | 2.0 | 52.0 |
| AVDC (Ko et al., 2023) (Flow) | 0.0 | 0.0 | 0.0 | 0.0 | 1.3 | 40.0 | 42.7 | 0.0 | 66.7 | 0.0 | 0.0 | 13.7 |
| AVDC (Ko et al., 2023) (Default) | 72.0 | 89.3 | 37.3 | 18.7 | 60.0 | 24.0 | 53.3 | 24.0 | 81.3 | 8.0 | 6.7 | 43.1 |
| LTM (Ranasinghe et al., 2025) | 77.3 | 95.0 | 39.0 | 20.3 | 82.7 | 84.3 | 52.3 | 68.3 | 98.0 | 10.3 | 7.7 | 57.7 |
| DAWN (ours) | 94.7 | 97.3 | 42.0 | 24.7 | 92.0 | 91.7 | 76.3 | 79.0 | 98.0 | 12.7 | 10.7 | 65.4 |

For each task, at most 360 action steps are performed unless the task is successfully completed prior to that. The success rate for each consecutive task is averaged across the 1000 episodes and reported. Considering the 5 continuous tasks as a sequences, the average number of tasks completed by the policy (i.e. average length) is also reported.

**Results:** We report results under two training settings, first without using any external robotic demonstration data in Table 1 and second with external robotic demonstration data (DROID) in Table 2. Since we follow evaluation protocol identical to (Hu et al., 2024; Ranasinghe et al., 2025), baseline results in our tables are directly borrowed from these prior works.

In Table 1, our DAWN achieves state-of-the-art results, highlighting the promise of pixel-motion based representations for complex robotic manipulation tasks. We also report results for two ablated variants of our method, containing only the low-level Action Expert. These results highlight the clear impact of pixel motions in achieving the strong results of our overall DAWN framework. Two example rollouts are presented in Figure A.1.

In the case of using external robotic demonstration data, direct comparison to prior work (where different approaches use different pretraining data) is less straightforward. We report results for our DAWN that is trained jointly on the DROID dataset and CALVIN ABC→D split. Our DAWN outperforms several recent works and performs competitively against SOTA methods VPP (Hu et al., 2024) and DreamVLA (Zhang et al., 2025). In Table 2, VPP benefits from significantly more videos (including 193k human manipulation trajectories, 179k robot manipulation trajectories, CALVIN, MetaWorld, and additional real-world datasets) in its pretraining compared to ours. Similarly, DreamVLA was first pretrained on a language-free split of the CALVIN and the full DROID dataset.

Overall, our DAWN achieves state-of-the-art performance on CALVIN benchmark, demonstrating the scalability as well as strong data efficiency of intermediate pixel-motion based VLA approaches.

## 4.3 META-WORLD EXPERIMENTS

We next evaluate DAWN on the MetaWorld (Yu et al., 2019) simulated environment containing a Sawyer robot arm. We focus on 11 challenging tasks constructed following (Ko et al., 2023; Ranasinghe et al., 2025) since the original benchmark is not language conditioned. We select this environment-tasks setting to enable direct comparison to closely related prior works (Ko et al., 2023; Wen et al., 2023; Xu et al., 2024; Ranasinghe et al., 2025) that also leverage pixel or point trajectories for robot manipulation tasks.

**Dataset:** We use the training split from (Ko et al., 2023; Ranasinghe et al., 2025) containing 165 actionless videos for Motion Director training and 220 task demonstrations across the 11 tasks for Action Expert training. All baseline experiments and results are identical to those reported in prior works AVDC (Ko et al., 2023) and LTM (Ranasinghe et al., 2025). The baselines "BC" refer to behaviour cloning, with BC-Scratch containing a ResNet initialized from scratch and BC-R3M using a ResNet initialized from R3M (Nair et al., 2022). All baselines are trained as described in (Ko et al., 2023; Ranasinghe et al., 2025).

**Results:** We report these results in Table 3. Our approach achieves clear performance improvements compared to prior works on this challenging benchmark. We particularly emphasize the improved *semantic understanding* of our DAWN framework: notice how DAWN achieves significantly better performance on visually similar but semantically dissimilar task pairs such as `open-door` vs `close-door`. We attribute this to our efficient latent diffusion formulation that enables scalable

Table 4: **Real-world single lift-and-place evaluation**. For each task, we evaluate 20 episodes with random initialization and report number of episodes of the following four cases: (i) successful lifts of the instructed object, (ii) successful placements of the instructed object (column Success, higher is better), (iii) lifts of an incorrect object, and (iv) placements of an incorrect object (column Wrong Obj.). Note that (iii) and (iv) are still classified as failures, though they differ from complete failures to grasp or place the object. We include these cases to provide a clearer understanding of the failure patterns.

| | Apple | | Avocado | | Banana | | Grape | | Kiwi | | Orange | |
|---|---|---|---|---|---|---|---|---|---|---|---|---|
| | Success ↑ (i)→(ii) | Wrong Obj. ↓ (iii)→(iv) | Success ↑ (i)→(ii) | Wrong Obj. ↓ (iii)→(iv) | Success ↑ (i)→(ii) | Wrong Obj. ↓ (iii)→(iv) | Success ↑ (i)→(ii) | Wrong Obj. ↓ (iii)→(iv) | Success ↑ (i)→(ii) | Wrong Obj. ↓ (iii)→(iv) | Success ↑ (i)→(ii) | Wrong Obj. ↓ (iii)→(iv) |
| Enhanced Diffusion Policy | 5→4 | 9→8 | 6→6 | 6→4 | 5→4 | 6→4 | 4→3 | 8→6 | 5→5 | 6→5 | 4→4 | 8→7 |
| $\pi_0$ (Black et al., 2024b) | 10→9 | 9→9 | 6→6 | 12→10 | 5→3 | 11→6 | 8→5 | 10→8 | 5→3 | 12→12 | 8→7 | 11→11 |
| DAWN | **12→10** | **2→2** | **10→10** | **1→1** | **9→7** | **0→0** | **15→10** | **1→0** | **13→11** | **0→0** | **11→11** | **2→2** |
| VPP (Hu et al., 2024) | 16→14 | 2→2 | 15→15 | 2→0 | 15→14 | 0→0 | 17→17 | 1→0 | 15→15 | 2→0 | 16→14 | 0→0 |
| DAWN* | **19→19** | **0→0** | **20→19** | **0→0** | **17→16** | **0→0** | **19→19** | **0→0** | **17→16** | **2→2** | **18→16** | **0→0** |

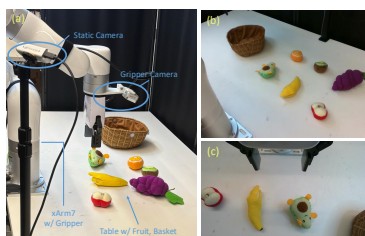

Figure 4: **Real world environment examples.** a) Our real-world environment includes a robot arm and two cameras. They are stereo RGB cameras, but we only use one RGB view from each camera. b) The RGB image from the static camera. c) The RGB image from the gripper camera.

Table 5: Ablation study on CALVIN dataset.

| Setting | Avg. Length |
|---|---|
| *(a) Pixel Motion vs RGB Goal* | |
| None | 2.78 |
| RGB Goal | 3.21 |
| Pixel Motion w/o pretrained | 3.42 |
| Pixel Motion | **4.00** |
| *(b) Gripper View* | |
| VPP w/o gripper view | 3.58 |
| DAWN w/o Gripper view | 3.74 |
| DAWN w/ Gripper View | **4.00** |
| *(c) # of Diffusion steps of Motion Director* | |
| 2 | 3.88 |
| 10 | 3.96 |
| 25 | **4.00** |
| 40 | 3.95 |

language-video pretraining, which in turn endows our model with stronger language understanding. We also highlight the improved performance in tasks such as `basketball` and `assembly`, which we attribute to our action-expert design choices that enable better robot state awareness.

We take these results as clear indication to how our design choices elevate the capabilities of intermediate pixel motion based VLA approaches, establishing the promise of this direction.

## 4.4 REAL WORLD EXPERIMENTS

We set up our real-world environment with a 7-DoF xArm7 robot arm and two RGB cameras: one providing a fixed third-person view from the right side of the arm, and the other mounted above the gripper (see Figure 4). A dataset of one thousand episodes is then collected, comprising lift-and-place manipulations involving six types of toys and a container.

**Implementation:** We compare our approach against three strong baselines. The first is our modification of Diffusion Policy (Chi et al., 2023), Enhanced Diffusion Policy, which is identical to our Action Expert but without pixel motion from a Motion Director. This model is pretrained on CALVIN ABC dataset. The second baseline is $\pi_0$ (Black et al., 2024b), where we initialize from the $\pi_0$ base model and apply Low-Rank Adaptation (LoRA) (Hu et al., 2022). The third is VPP Hu et al. (2024), initialized from their official pretrained checkpoint. We also build a variant of DAWN (DAWN*) that can benefit from the VPP pretrained checkpoint (details in Appendix C) to enable fair comparison with VPP. All methods are fine-tuned on our collected real-world dataset for 100k steps. The task is highly challenging for the policy to learn, as a total of only 1k episodes across 12 tasks provides very limited training data.

**Evaluation:** We evaluate all methods using the lift-and-place task pair with different objects, where the robot is instructed to lift a specified object and place it into a container. We record the number of episodes in which the robot: (i) successfully lifts the correct object, (ii) successfully places the correct object, (iii) lifts an incorrect object at the end with 500 max steps, and (iv) places the incorrect object from the previous lifting. Note that (iii) and (iv) are still classified as failures, though they differ from complete failures to grasp or place the object. We include these cases to provide a clearer understanding of the failure patterns. Each episode begins from a randomly initialized environment, and we run 20 episodes per task in total.

Results in Table 4 demonstrate that our method achieves higher success in lifting and placing the correct object compared to the baselines, despite using far fewer parameters than $\pi_0$. In contrast, without the pixel motion provided by Motion Director, our Enhanced Diffusion Policy baseline

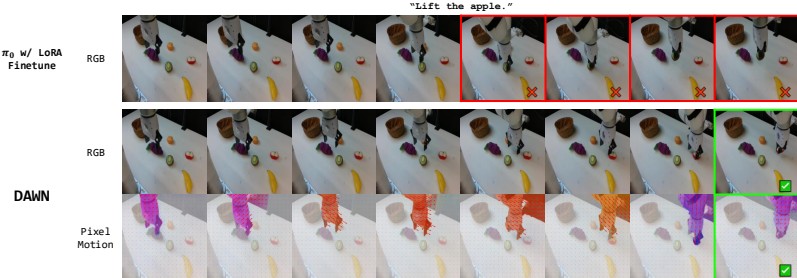

Figure 5: **Real world rollout examples**. Given a task of "lift the apple", the first row shows the rollout image sequence by $\pi_0$ with LoRA finetuned, which lifts the wrong object, kiwi. The second row shows a successful episode by our method in the same environment setting, and the third row is the visualization of the corresponding pixel motions predicted by Motion Director.

frequently fails by lifting the wrong object or completely failing the task. Compared to $\pi_0$, DAWN exhibits better semantic awareness, lifting the correct object more often. Figure 5 shows an example episode where $\pi_0$ fails to follow the instruction and lifts the wrong object, and DAWN can lift the correct object. DAWN is both more accurate and more parameter-efficient than the baselines in this set of challenging tasks.

The VPP baseline requires significantly more pretraining than our setup to perform well on our real world tasks (see Appendix C). Interestingly, our similarly trained DAWN* variant consistently outperforms VPP, demonstrating that structured pixel motion provides complementary benefits and can further strengthen even strong two-stage diffusion based methods such as VPP.

### 4.5 ABLATION STUDY

We conduct ablation experiments on the CALVIN ABC→D benchmark to assess the impact of (a) structured pixel motion representation, (b) gripper view conditioning, and (c) the number of diffusion steps (See Table 5).

**(a) Pixel Motion, RGB Goal, and pretraining.** We compare two variants against our default setting: (i) RGB goal image conditioning instead of pixel motion; (ii) Only Action Expert w/o pixel motion, and (iii) generating pixel motion with a denoising U-Net trained from scratch. As shown in Table 5(a), pixel motion yields the best performance, highlighting its utility as a structured and interpretable intermediate, and our method benefits a lot from the pretrained image generation model, even though the model was not trained for pixel motion generation before.

**(b) Gripper View Conditioning.** We further ablate the effect of adding egocentric gripper-mounted observation to Motion Director. Removing the gripper view leads to performance degradation (3.74 vs. 4.00), while prior methods such as VPP degrade further (3.58). These results confirm that the additional viewpoint facilitates reasoning about occlusions and fine-grained hand-object interactions.

**(c) Diffusion Steps of Motion Director.** Motion Director module can capture meaningful motion information even at 2 diffusion steps (3.88). Increasing the number of steps steadily improves performance, peaking at 25 (4.00) and can't gain more beyond that (e.g., 40 steps with 3.95).

## 5 CONCLUSION

In this work, we present a two-stage diffusion-based visuomotor framework for robot manipulation, termed DAWN, which achieves state-of-the-art performance on CALVIN, MetaWorld, and real-world benchmarks. Instead of using manifest visual information in RGB space, we use explicit dense pixel motion representations as a structured interface between a latent diffusion Motion Director and a diffusion-based Action Expert. This design bridges hierarchical motion decomposition and end-to-end agents while preserving interpretability and modularity. By instantiating both stages with modern diffusion models and leveraging strong pre-trained vision language backbones, DAWN delivers high data efficiency and robust transfer, indicating that much of the gap between multi-stage tracking pipelines and VLA/latent-feature hierarchies stems not from the framework itself but from underpowered high- and low-level components. We hope DAWN encourages re-examining structured intermediate representations as a practical path to interpretable, data-efficient robot control.

## REPRODUCIBILITY STATEMENT

Our code and models will be released publicly. All datasets used for our pretraining are publicly available data. The real world robot demonstration dataset used for fine-tuning will be publicly released. All evaluations reported in our paper follow code implementations from prior published work that are publicly available.

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

# Appendix

## A    TRAINING DETAILS

**Training Details.**    We train all models on 4 NVIDIA A6000 GPUs. For Motion Director, we train for 100k iterations with a per-GPU batch size of 16. For Action Expert, we train for 10k iterations with a per-GPU batch size of 64. We use the AdamW optimizer with a learning rate of $1 \times 10^{-4}$. Mixed precision training is used to reduce memory usage and improve throughput. All training is implemented in PyTorch with the HuggingFace Diffusers and Transformers libraries.

## B    DATASET DETAILS

### B.1    CALVIN

CALVIN is an open source simulated benchmark to learn long-horizon language-conditioned tasks, which contains 4 different simulation environments-A, B, C, D. While each split (A–D) shares the same robotic setup, variations in object placement, textures, lighting, and distractors ensure that models cannot rely on memorization but must instead demonstrate robust visuomotor understanding. The 34 manipulation tasks span a wide range of skills such as pushing, placing, rotating, toggling switches, and opening drawers, all expressed through natural language instructions.

In our approach, we adopt a hierarchical inference strategy where Motion Director predicts a pixel motion plan, and Action Expert executes this plan by directly applying 10 consecutive low-level action steps before requesting a new pixel motion. This design reduces the computational overhead of repeatedly invoking the diffusion-based planner, while ensuring that each high-level motion is translated into a temporally coherent sequence of actions. Our two example rollouts are presented in Figure A.1. These frame sequences show some intermediate steps' observations from the static camera view, and the pixel motion plans visualized with the observations, which align and guide the action steps from a high-level guidance.

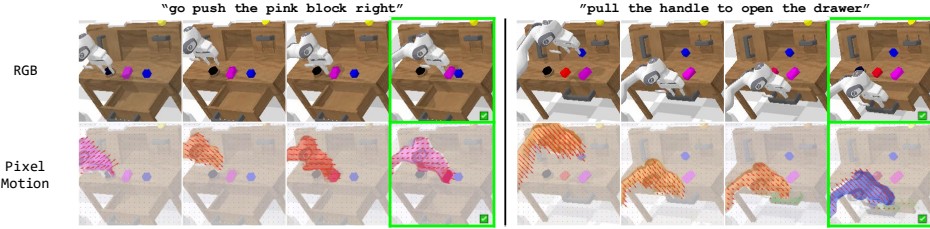

Figure A.1: **CALVIN rollout examples**. Two example rollouts of DAWN in CALVIN environment. The first row is the sequence of RGB images, and the second row is the visualization of the corresponding pixel motions predicted by Motion Director.

### B.2    DROID

DROID is a large-scale "in-the-wild" robot manipulation dataset featuring 76k real demonstration trajectories across 564 varied scenes and 86 tasks. It provides over 350 hours of interaction data, with diverse viewpoints, object types, and natural instruction annotations.

### B.3    REALWORLD

We constructed a dataset specifically for fine-tuning and real-world evaluation. The experimental platform consists of a 7-DoF xArm7 manipulator and two RGB cameras. An Intel RealSense D435 was positioned laterally to provide a third-person view of the workspace, while an Intel RealSense D405 was mounted above the gripper to capture a close-up view of the end-effector and its interactions with objects. Though both cameras are stereo cameras, we only use a single RGB view from

Table A.1: **Comparison with VPP in Real world experiment..**

| | Apple | | Avocado | | Banana | | Grape | | Kiwi | | Orange | |
|---|---|---|---|---|---|---|---|---|---|---|---|---|
| | Success | Wrong Obj. | Success | Wrong Obj. | Success | Wrong Obj. | Success | Wrong Obj. | Success | Wrong Obj. | Success | Wrong Obj. |
| VPP (Hu et al., 2024) | 16→14 | 2→2 | 15→15 | 2→0 | 15→14 | 0→0 | 17→17 | 1→0 | 15→15 | 2→0 | 16→14 | 0→0 |
| DAWN * | **19→19** | **0→0** | **20→19** | **0→0** | **17→16** | **0→0** | **19→19** | **0→0** | **17→16** | 2→2 | **18→16** | **0→0** |

each camera in all the experiments. This dual-camera setup enables complementary perspectives, facilitating both scene-level and fine-grained observations.

Data was collected through a leader–follower teleoperation scheme, where a human operator controlled a leader device to guide the motions of the xArm7 (follower). Each demonstration episode was restricted to a single atomic task, such as lifting a fruit, transporting it, or placing it into a basket. Episodes were initialized either from randomized joint configurations or from the terminal state of the preceding task, ensuring diversity in initial conditions. To further increase variability and promote generalization, we occasionally re-dropped and re-grasped objects within the same episode.

The resulting dataset comprises 1,000 episodes, with a minimum of 100 demonstrations allocated to each distinct task. This distribution ensures both task balance and sufficient coverage for downstream fine-tuning. Overall, the dataset provides a structured yet diverse collection of manipulation trajectories suitable for evaluating task-specific policies under realistic conditions.

## C   DAWN VARIANTS

In this section, we describe the DAWN variant reported as DAWN* in Table 4 (repeated here in Table A.1). VPP (Hu et al., 2024) provides a strong pretrained checkpoint for a diffusion backbone that is trained on large-scale robotic demonstration datasets. Since VPP does not perform at its highest level when trained on less data (e.g. only our real world dataset), we adopt its pretrained official checkpoint for the real world evaluations. To enable fair comparisons with this model, pretraining on a similar scale of robotics demonstration data is beyond our compute capacity. Therein, we adopt a variant of DAWN that can use the VPP pretrained checkpoint and we finetune it to generate intermediate pixel motion representations. Since VPP is originally trained for generating RGB representations, we simply generate Pixel Motion representations in addition to RGB to benefit from the pretraining. Both these features, RGB and Pixel Motion, are subsequently provided to the Action Expert module. For fair comparison, we use the same Action Expert as VPP. Secondly, we also limit the reverse diffusion iterations of Motion Director to 1 (instead of our default 25) for fair comparison, since VPP follows the same setting.

These results (repeated here in Table A.1) from evaluation under identical settings establish how our proposed structured pixel motion can further improve upon even a strong diffusion based approach such as VPP.

