# OpenReview forum: "Pixel Motion Diffusion is What We Need for Robot Control"
_ICLR.cc/2026/Conference — ICLR 2026 Conference Withdrawn Submission_

### Official Review · Reviewer_eNMy · 2025-10-26

**Soundness:** 3
**Presentation:** 3
**Contribution:** 2
**Rating:** 4
**Confidence:** 5

**Summary:**

This paper proposes a bi-level policy for robotic manipulation, where the top predicts 2D pixel motions and the bottom conditioned on pixel motions predicts the robot's end-effector control.  In contrast to LangToMo, thks paper proposes to build the pixel motion generator atop more performant 2D diffusion models--a latent diffusion model (LDM) pre-trained on internet-scale image datasets.  Therefore, this paper achieves state-of-the-art performance on both CALVIN and MetaWorld simulation benchmark.

**Strengths:**

1. This paper is clearly written.  It's easy to follow the implementation detail.
2. The proposed method achieves state-of-the-art performance on both CALVIN and MetaWorld simulation benchmark.

**Weaknesses:**

1. This paper does not discuss the latency / computational overhead imposed by latent diffusion models. Since deployment of robot policies often requires computing on edge devices, it's critical to understand the speed and GPU memory needed during inference. I'd highly encourage the authors to include these numbers in Table 1 and 2.
2. This paper lacks a strong baseline--MoDE [1], in Table 1.  The baseline achieves 4.01 using additional datasets.
3. The core idea of this paper closely follows LangToMo, while the only difference is the choice of pixel motion generator: DAWN adopts LDM while LangToMo uses explicit pixel diffusion models.  I'd like to argue this is not novelty but a better design choice.
4. This paper lacks discussion a very strong baseline published in CoRL 2025--FLOWER [2], although comparison against the model is not needed.
5. This paper only demonstrates on two simulation benchmark, especially one (MetaWorld) is a toy-ish benchmark.  I'd highly encourage the authors to follow FLOWER that evaluates on CALVIN, LIBERO, SIMPLERENV, ALOHA SIM and Kitchen simulation benchmark.

---

Reference:

[1] M. Reuss, J. Pari, P. Agrawal, and R. Lioutikov. Efficient diffusion transformer policies with mixture of expert denoisers for multitask learning, 2024

[2] Reuss, Moritz, et al. "Flower: Democratizing generalist robot policies with efficient vision-language-action flow policies." arXiv preprint arXiv:2509.04996 (2025).

**Questions:**

1. What is the latency and required GPU memory during inference?
2. What is the performance of DAWN on LIBERO and SIMPLEREnv (and possibly ALOHA SIM and Kitchen) simulation benchmark?

---

### Official Review · Reviewer_ExPK · 2025-10-29

**Soundness:** 3
**Presentation:** 3
**Contribution:** 2
**Rating:** 2
**Confidence:** 3

**Summary:**

The paper “Pixel Motion Diffusion Is What We Need for Robot Control” proposes DAWN, a two-stage diffusion framework that uses pixel motion as an explicit and interpretable intermediate representation between vision-language inputs and robot actions. The first stage, Motion Director, generates dense pixel-motion fields conditioned on multi-view observations and language instructions, while the second stage, Action Expert, translates these motions into low-level robot actions via another diffusion model. This modular design combines the scalability of diffusion models with the interpretability of motion representations. DAWN achieves state-of-the-art results on CALVIN and MetaWorld benchmarks and performs competitively on real-world xArm7 tasks.

**Strengths:**

- Thorough evaluation across two major simulation benchmarks (CALVIN and MetaWorld) and real-world robot experiments, demonstrating both scalability and practicality.

- Clear and intuitive figure illustration that effectively conveys the proposed two-stage diffusion framework and its intermediate pixel-motion representation.

- The model offers strong interpretability, as the explicit pixel-motion predictions make the underlying motion planning process visually understandable and analysable.

**Weaknesses:**

- The distinction from previous work is not clearly articulated. The writing around L52–53 does not sufficiently highlight the unique scientific problem addressed by the proposed two-stage, pixel-motion-based framework, for it just claims that the model design is out-of-dated. As a result, it remains difficult to discern how this approach **fundamentally** differs from prior methods such as LangToMo. It is sincerely suggested to include a comparative figure or schematic in the introduction to visually illustrate the architectural and conceptual differences, thereby better emphasizing the novelty and insight of this work.

- The manipulation setting appears somewhat constrained. The proposed framework relies on a fixed third-person camera to fully observe the workspace and end-effector, which limits its applicability to industrial setups where the robot is mounted on a stationary table. In contrast, for humanoid or mobile robotic scenarios—where external cameras may move or be absent—the method’s effectiveness is unclear. It would strengthen the paper to clarify this limitation or to experimentally evaluate the impact of camera motion and reduced observability.

- The real-world evaluation tasks are relatively simple. The demonstrated pick-and-place tasks may not fully showcase the advantages of pixel-level motion planning. More complex manipulation tasks, such as pouring, folding, or tool use, would better highlight the method’s generality and robustness.

- The benchmark choice could be more up-to-date. Both CALVIN and MetaWorld are now considered somewhat dated. Evaluating on newer or more diverse benchmarks, such as RoboSuite or RoboWin, would further support the validity and modern relevance of the proposed approach.

If my concerns are adequately addressed, particularly W1 and W2, I would be willing to reconsider and potentially raise my overall score.

**Questions:**

- The description of Lang2Mo around L71 is unclear. It is not well explained what “pixel-level diffusion” specifically refers to in that context—is it means diffusion to generate next frame? Furthermore, it remains ambiguous why the resolution of the generated motion representation and the training capability of pixel-level diffusion models are said to be limited. Providing a more precise explanation of what “pixel-level diffusion” entails, how it differs from latent diffusion in this work, and why resolution or training constraints arise would help readers better understand the motivation for your design choice.
- The paper would benefit from a clearer discussion of computational efficiency and real-time feasibility, as diffusion models often entail high inference latency and your framework works in a cascaded manner that may hinder deployment in closed-loop control.

---

### Official Review · Reviewer_ztqs · 2025-11-01

**Soundness:** 2
**Presentation:** 3
**Contribution:** 2
**Rating:** 4
**Confidence:** 3

**Summary:**

This paper introduces DAWN (Diffusion is All We Need), a two-stage diffusion-based framework for language-conditioned robotic manipulation. The approach uses a Motion Director (latent diffusion model) to generate pixel motion representations from visual observations and language instructions, which are then consumed by an Action Expert (diffusion policy) to produce executable robot actions. The method is evaluated on CALVIN, MetaWorld, and real-world manipulation tasks, achieving competitive or state-of-the-art results despite using limited data and smaller model capacity compared to recent VLA models.

**Strengths:**

- The use of explicit pixel motion as an intermediate representation provides interpretability while maintaining end-to-end trainability.
- The paper includes extensive experiments across simulation and real-world settings, with reasonable ablation studies.
- The framework demonstrates strong performance with substantially less data and smaller model capacity than competing VLA models, which is practically valuable.

**Weaknesses:**

- The contribution is limited. Both latent diffusion models and diffusion policies are well-established techniques. The main contribution seems to be combining these with pixel motion as an intermediate representation.
- The relationship to VPP (Hu et al., 2024) needs clearer differentiation. Both use video diffusion and action policies, but VPP operates in latent RGB embedding space while DAWN uses explicit pixel motion.
- The paper doesn't discuss or compare inference time or computational requirements. Two separate diffusion processes could be computationally expensive.
- The paper doesn’t provide many qualitative results between predicted motion and the ground truth, thus remaining unclear how well the motion director could help. Moreover, the ground truth depends on the RAFT predictions, which might have noise.
- In real-world scenarios, motion blur, occlusions, or lighting changes could affect the motion. The real-world motion has a distribution gap with the simulation motion.
- The paper lacks direct comparison with LangToMo (Ranasinghe et al., 2025), which also uses pixel motion but with pixel-level diffusion. Also the authors do not discuss why latent diffusion is superior to pixel-level diffusion.

**Questions:**

- What if we discard the intermediate pixel motion output and train an end-to-end policy, and use the motion prediction as one term of the objectives?

Also see weakness.

---

### Official Review · Reviewer_C8Qf · 2025-11-02

**Soundness:** 2
**Presentation:** 3
**Contribution:** 2
**Rating:** 4
**Confidence:** 4

**Summary:**

This paper presents a two-stage training framework aimed at optimizing a robot control policy. Specifically, in the first stage, the authors utilize inter-frame optical flow to generate pixel motion images, which provide intuitive motion information for subsequent policy learning. In the second stage, these generated pixel motion images serve as additional conditional inputs to enhance the predictive capabilities of the policy. The authors conducted thorough experiments in both simulated environments and on real robot platforms. The results validate the effectiveness and generality of the core idea of this approach. This method improves the accuracy and robustness of robot control.

**Strengths:**

This paper aims to improve robot control through a two-stage training method that utilizes pixel motion. This research is crucial because effectively using video data to enhance the learning efficiency of robots has been a long-standing challenge. The authors highlight that employing diffusion models for generative tasks can result in improved outcomes.

**Weaknesses:**

While pixel motion is certainly a topic worth exploring, I have several concerns:

1. The abstract does not clearly define the specific problem that the paper addresses, and the Hyperlink in the abstract is inaccessible.
2. There are concerns regarding the novelty of the work. Similar studies have been conducted in the past, and although the authors reference related research, they do not sufficiently highlight the differences. It is essential to emphasize these distinctions in order to clearly demonstrate the contributions of this paper.

[1] Any-point Trajectory Modeling for Policy Learning
[2] Flow as the Cross-Domain Manipulation Interface

**Questions:**

1. There is a lack of evaluation criteria to support the generative performance of stage one. Since optical flow is highly sensitive to lighting conditions, the experimental setup—specifically the lighting environments and the visibility of the robotic arm—was relatively controlled. If the static camera were positioned in a more open environment, would the accuracy of optical flow labeling still be reliable under those conditions?
2. The primary contribution appears to be in the initial stage. Since similar studies focus on this generative phase, could more explicit metrics be provided to demonstrate that using diffusion and optical flow for labeling yields better results compared to the initial stages of other methods?
3. The authors should specify the parameters used for comparisons with ATM and im2Flow2Act to ensure fair evaluation. Otherwise, the comparisons may seem biased.
4. What is the execution efficiency? The work uses diffusion for both stages, but due to diffusion's inherent nature, achieving real-time performance could be significantly challenging. More detailed inference information needs to be provided.

---

### Author Response · Authors · 2025-11-14
**Response to the review**

We thank all the reviewers for their insightful comments and suggestions. We would like to first clarify some shared misunderstandings and then provide more information to address the reviewers’ concerns.

## 1. **The Novelty of the paper**
To address the reviewers’ questions on novelty, we summarize the key distinctions between our approach and closely related concurrent or prior works.
### (a) Comparison to LangToMo (LTM)
While both approaches use pixel motion as an intermediate representation, our framework differs not just in design choices, but in several fundamental aspects:

- Latent diffusion for scalable, data-efficient motion generation: LTM performs diffusion in pixel space, which constrains resolution and requires pretraining on robot-demonstration datasets (e.g., OpenX) to achieve strong performance. Our Motion Director operates in the latent space of a pretrained diffusion model, enabling stable training, higher fidelity, and strong performance without large robot-specific pretraining.
- No reliance on externally computed motion cues at inference: LTM requires previous optical-flow frames as input to both modules. Our method predicts motion directly from the current observation, making it simpler, more general, and more suitable for closed-loop control.
- Stronger and more expressive action policy: LTM employs a small ViT-based policy head, whereas our Action Expert is a diffusion transformer, capable of capturing long-horizon dependencies and complex action distributions.

These differences lead to a more scalable, more expressive, and more modular design than LTM, even though both share the high-level intuition of using pixel motion.
### (b) Comparison to VPP

Our method also differs significantly from predictive-representation approaches such as VPP:

- Explicit motion vs. implicit predictive features: VPP treats a video diffusion model as a **feature extractor**, producing **predictive RGB representations** in a single forward pass during inference. In contrast, our Motion Director predicts a structured **pixel motion explicitly**. This places our method at a distinct point in the design space: explicit pixel motion rather than RGB-based predictive representation.

## 2. **Execution Efficiency**

We provide the following average inference time comparison (per control step) in our real-world experiment:

| Method | Inference Efficiency (ms) ↓ |
| :---- | :---- |
| Enhanced DP | 112.77 |
| π₀ | 571.89 |
| VPP | 190.55 |
| DAWN | 319.82 |


Enhanced DP is the fastest baseline since it uses a single diffusion model without any additional high-level module. VPP is slightly slower but remains efficient because its video diffusion model operates in a single forward pass to extract predictive features, without explicitly generating video frames or pixel-level outputs. While DAWN takes longer to compute due to its two-stage design, our method still runs at practical closed-loop frequencies. It achieves the highest overall success rates, indicating that the benefits of explicit motion reasoning outweigh the added inference time.

## 3. **Pixel Motion Generation Performance**
We include additional qualitative comparisons between predicted pixel motion and RAFT-derived ground truth to illustrate the quality of the Motion Director’s outputs at: (https://drive.google.com/file/d/1stsI_jA24QyIny8k1hRuK66xmRDD8v_C).

To further validate prediction quality, our visualization page (https://anonymous.4open.science/w/DAWN) provides multiple real-world and CALVIN inference sequences showing predicted pixel motion alongside the executed robot behavior. Across these examples, the predicted motion fields are clean, task-aligned, and remain stable throughout closed-loop control.
Importantly, these explicit pixel-motion maps make the intended motion of objects and the end-effector visually interpretable, providing a clear and actionable intermediate representation that directly guides the downstream policy.

## 4. **The reliability of RAFT or optical flow labeling**

Reviewers raised concerns about the sensitivity of optical flow(OF) to lighting conditions or experimental setup or RAFT’s OF quality.

We clarify that:

- RAFT is used **only during training** as supervision for pixel motion.
- The Motion Director does **not** rely on RAFT at **inference time**.
- All the challenges could be addressed by simply switching to a better OF algorithm, which is not the research focus of this paper.

## 5. **Why latent diffusion instead of pixel-level diffusion**

- Latent diffusion allows us to directly fine-tune a pretrained generative prior (Stable Diffusion v1.5), enabling data-efficient training and high-quality motion generation. Pixel-level diffusion approaches require large-scale pretraining (e.g., OpenX for LangToMo) before fine-tuning.
- Operating in latent space provides **compute-efficient, high-resolution outputs**, allowing **dense pixel motion** prediction while keeping the denoising process lightweight.

---

### Note · Authors · 2025-11-14

I have read and agree with the venue's withdrawal policy on behalf of myself and my co-authors.